# Solvent-Free Procedure for the Preparation under Controlled Atmosphere Conditions of Phase-Segregated Thermoplastic Polymer Electrolytes

**DOI:** 10.3390/polym11030406

**Published:** 2019-03-01

**Authors:** Álvaro Miguel, Francisco González, Víctor Gregorio, Nuria García, Pilar Tiemblo

**Affiliations:** Instituto de Ciencia y Tecnología de Polímeros, ICTP-CSIC, Juan de la Cierva, 3, 28006 Madrid, Spain; amiguel@ictp.csic.es (Á.M.); fgonzalez@ictp.csic.es (F.G.); v.gregorio@ictp.csic.es (V.G.)

**Keywords:** polymer electrolytes, thermoplastic electrolytes, solid electrolytes, ionic liquids, energy storage

## Abstract

A solvent-free method that allows thermoplastic solid electrolytes based on poly(ethylene oxide) PEO to be obtained under controlled atmosphere conditions is presented. This method comprises two steps, the first one being the melt compounding of PEO with a filler, able to physically crosslink the polymer and its pelletizing, and the second the pellets’ swelling with an electroactive liquid phase. This method is an adaptation of the step described in previous publications of the preparation of thermoplastic electrolytes by a single melt compounding. In comparison to the single step extrusion methodology, this new method permits employing electroactive species that are very sensitive to atmospheric conditions. The two-step method can also be designed to produce controlled phase-segregated morphologies in the electrolyte, namely polymer-poor and polymer-rich phases, with the aim of increasing ionic conductivity over that of homogeneous electrolytes. An evaluation of the characteristics of the electrolytes prepared by single and two-step procedures is done by comparing membranes prepared by both methods using PEO as a polymeric scaffold and a solution of the room-temperature ionic liquid 1-ethyl-3-methylimidazolium bis(trifluoromethanesulfonyl) imide (EMI TFSI) and the bis(trifluoromethanesulfonyl) imide lithium salt (Li TFSI) as liquid phase. The electrolytes prepared by both methods have been characterized by Fourier transform infrared spectroscopy and optic microscopy profilometry, differential scanning calorimetry, self-creep experiments, and dielectric spectroscopy. In this way, the phase separation, rheology, and ionic conductivity are studied and compared. It is striking how the electrolytes prepared with this new method maintain their solid-like behavior even at 90 °C. Compared to the single step method, the two-step method produces electrolytes with a phase-separated morphology, which results in higher ionic conductivity.

## 1. Introduction

Batteries are a ubiquitous item in everyday life, as a fundamental part of portable devices or as a basic resource in the exploitation of intermittent energy sources. Each application has its own requirements, in some cases, very high energy density, in others, small size and weight, or thin and flexible geometries, but all of them have in common the need to fulfill very strict safety standards, mainly referring to toxic leaks and short circuits arising from dendrite growth. This second issue has in fact slowed down the extensive use of secondary Li metal batteries, which, however, have excellent energy density values. Not only Li, but also other cations such as Al and Na produce dendrites growing from the anode, which may short circuit the battery, and produce explosions and fire in the case of the battery containing inflammable components [1,2]. Both leaks and dendrite growth are related to the use of liquid electrolytes. Dendrites, which are at the origin of the most important safety issues, would be far less probable in solid electrolytes, especially when the elastic modulus of the solid is high enough [1].

It is becoming progressively more evident that future batteries will comprise a solid and not a liquid electrolyte because of safety [3,4], and drawbacks of solid electrolytes, such as bad interfacial contact and low conductivity, i.e., low ion diffusivity, will have to be dealt with. Both organic (polymeric) and inorganic (ceramic) electrolytes exist, each one with its own flaws and advantages. Polymers are electrochemically and thermally more unstable than ceramics, but their lightness, flexibility, and toughness are incomparable, and interfacial contact may be very good [5,6]. Ion conductivity can become roughly the same in both ceramic- and polymer-based electrolytes. There exist polymers which themselves are cationic conductors, namely different polyanions, but currently the most important polymeric cation conductors are polymers containing sequences of oxyethylenic units, in particular polyethylene oxide (PEO), which is the best known and most widely used polymer in Li electrolytes. Polyether sequences are able to complexify cations and transport them by the segmental motions of the polymer’s amorphous phase. PEO by itself is a poor cation conductor compared to liquids, not only because of the existence of the crystalline phase (which does not conduct) but also because of the very high viscosity of the polymer’s amorphous phase, even at temperatures as high as 70 °C. This is why PEO-based solid electrolytes include high fractions of liquid phases [5,6,7,8]. There is now extensive knowledge on different techniques, allowing for the incorporation of liquid phases into PEO, and numerous reviews give accounts of them [1,7,8].

The main strategies are the soaking of a polymer mat or porous polymer membrane in the electroactive liquid phase, or the preparation of chemically or physically crosslinked gels or blends, which are dense and not porous membranes. Porous soaked membranes may not fully address safety requirements, as it has been proven [1] that not all dimensionally stable polymer-based electrolytes manage to completely eliminate dendrite growth. It is easy to see that if at a microscopic level, where the electrolyte is very much like a liquid in sufficiently large regions, then dendrites will grow. Both dendrite growth and cationic transport are governed by ion diffusivity and so there is a trade-off between cation conductivity and dendrite growth. Electrolytes that completely avoid dendrite growth because of their high elastic modulus display conductivities so low as to make them uninteresting. A way to circumvent this trade-off is the design of polymer-based dense (as opposed to porous or tissue-like) electrolyte membranes with controlled liquid-rich and solid-rich phase separation, where the high elastic modulus solid-rich phase is able to stop dendrite growth efficiently, while the liquid-rich phase is able to efficiently transport ions. This strategy is not only valid for Li cations but for other metal cations where dendrite growth is also an issue, such as Na, Al, Mg [9], and Zn [10,11] cations.

For the last few years, we have explored the preparation, with solvent-free procedures, of thermoplastic physically crosslinked electrolytes for Li ion transport, based on PEO, which has ionic conductivities (σ) well over 0.1 mS cm^−1^ at 25 °C. These thermoplastic electrolytes have been prepared by melt compounding all the components of the electrolyte in a single step, employing lithium salts as an electroactive liquid phase, dissolved in both conventional carbonates [12,13] and room temperature ionic liquids (RTIL) [14,15]. The use of an ad-hoc modified sepiolite, TPGS-S, as a physical crosslinking site for the PEO chains [12,13,14,15] is a key point in these materials. Physical, reversible crosslinking (as opposed to chemical, irreversible) makes these electrolytes solid-like up to 90 °C, but still thermoplastic, i.e., recyclable, reshapeable, and extrudable. These electrolytes, which are solid dense membranes, are characterized by no dendrite growth and good capacity cycling [16,17]. The electrolytes containing cyclic carbonates show σ of 0.8 mS cm^−1^ at 25 °C, but those incorporating imidazolium or pyrrolidinium ionic liquids are (for the same liquid fraction) about half that value. This is because of the higher viscosity displayed by the dissolution of Li salts in RTIL as compared to cyclic carbonates, and also because the excellent compatibility of pyrrolidinium and imidazolium liquids with PEO makes these electrolytes better blended than those prepared with carbonates. In effect, while in RTIL-based electrolytes, no phase separation is seen, in those prepared with cyclic carbonates, a microphase separation is detected [12]. This phase-separated morphology turns out to be then better adapted for efficient ion transport than a more homogeneous morphology. This observation has also been done by other authors before in the case of PEO-based electrolytes [18]. The adequacy of a controlled phase separation structure for ionic transport has also been shown in recent work on gel electrolytes based on polycationic scaffolds and ionic liquid electroactive phases [19,20], in which the existence of phase separation leads to solid materials with very large ion diffusion coefficients and conductivities.

Forcing a microphase separation in the RTIL/PEO thermoplastic electrolytes mentioned above [14,15,16,17] has been the aim of our recent work. At the same time, and because of our interest in electrolytes for aluminium ion transport, we have made efforts to adapt the melt compounding procedure so that it will allow for producing thermoplastic electrolytes with electroactive liquid phases, which are very sensitive to humidity. In this work we present a two-step procedure for the preparation of thermoplastic electrolytes with controlled phase-separated morphology, which is solvent-free and scalable, and which allows humidity sensitive electroactive phases to be used.

## 2. Materials and Methods 

### 2.1. Materials

All the materials were used as received, unless otherwise indicated. Poly(ethylene oxide) 5·10^6^ g mol^−1^ molecular weight (*M*_w_), Bis(trifluoromethane) sulfonimide lithium salt (LiTFSI) with a purity of 99.95%, and D-α-tocopherol polyethylene glycol 1000 succinate (TPGS) were purchased from Sigma–Aldrich (Madrid, Spain). 1-Ethyl-3-Methylimidazolium Bis(trifluoromethanesulfonyl) imide (EMITFSI) with a purity of 99.5% was purchased from Solvionic (Toulouse, France). Neat sepiolite was kindly supplied by Tolsa. Sepiolite surface modification (Madrid, Spain) with TPGS to render TPGS-S was carried out as described in [21].

### 2.2. Preparation of the Electrolytes

The two-step method for the preparation of electrolytes described below is based on the single-step melt compounding method described elsewhere [14,15]. To compare the outcome of both methods with regards to morphology, mechanical stability, rheology, and ionic conductivity, two electrolytes with the same formulation were prepared in both ways.

*One-step procedure*. In this procedure, the components were melt-compounded together in a Haake MiniLab extruder at 120 °C, 80 rpm, in 20 min residence time. Detailed description of this method can be found in [12,13,14,15]. This procedure was used to prepare electrolyte EMITFSI-1 in Table 1.

*Two-step procedure*. In Step 1, a 5 wt % of modified sepiolite (TPGS-S) is physically mixed with PEO in a mortar. This mixture is melt compounded in a Haake MiniLab extruder (Thermo Fischer Scientific, Walthman, MA, USA) at 120 °C and 80 rpm, in about 20 s residence time. The extrudate is then cut into pellets of about 3 mm in size. In Step 2, 45 mmol of the 3 mm pellets are placed on a small glass crystallizer and put into a glass oven, Büchi B-585, at 80 °C. With the help of an ultrasonic bath Branson 1210 (Marshal Scientific), 2.44 mmol of LiTFSI are dissolved in 4.6 mmol of EMITFSI for 20 min. Then, with a Pasteur pipette, the adequate amount of solution is poured all over the 3 mm pieces to cover them all, after which the mixture is left overnight in the oven at 80 °C. This time lapse is sufficient to have all of the liquid phase absorbed by the PEO/TPGS-S pellets. It is carefully checked that all pellets are similarly swollen to the naked eye and that none remain unchanged. It is also checked that no remaining liquid phase exists. This procedure was used to prepare the electrolyte EMITFSI-2 in Table 1. 

This soaking procedure was followed by different-sized PEO/TPGS-S particles, from powdered PEO/TPGS to actual 2 cm-diameter PEO/TPGS-S whole membranes. The 3 mm-sized pellet is the best suited sample format for a complete swelling of the particles and the complete absorption of the liquid phase. This is so because the smaller the size, the more efficient the swelling of the liquid by the polymeric phase, due to the volume/surface ratio of the pellets. For the polymer/liquid ratio employed in this work, pellets significantly smaller than 3 mm are very difficult to swell homogeneously, and some of them remain dry while others become highly swollen. On the other hand, if very large pellets are used, some liquid phase may remain unused.

In order to understand the role of TPGS-S in the final properties of the two-step electrolyte EMITFSI-2, an electrolyte like EMITFSI-2 but without TPGS-S was prepared and appears in Table 1 as EMITFSI-2a. Finally, EMITFSI-2b, without TPGS-S and LiTFSI, was prepared so as to understand the contribution of the Li salt to the mechanical properties of these thermoplastic electrolytes. Scheme 1 illustrates both of these procedures. 

*Film preparation*: For conductivity measurements, calorimetry, profilometry, and infrared spectroscopy, 500 µm films were made by hot press molding with a Specac press at 75 °C and 2 Tn for 3 min. All the membranes in Table 1 are self-standing and easy-to-handle solids at room temperature.

### 2.3. Characterization

ATR–FTIR: IR spectra were recorded on the surface of the electrolytes using an FTIR PerkinElmer Spectrum Two (Perkin Elmer, Walthman, MA, USA), with four scans and resolution 1 cm^−1^. On each film, several spectra were recorded to study their structural heterogeneity.

Differential scanning calorimetry (DSC) studies were performed in a TA instrument Q100 (TA Instruments, New Castle, DE, USA). The heat flow was recorded as follows: Two cooling–heating cycles at 10 °C min^−1^ from 120 to −60 °C and again to 120 °C, followed by a third cooling–heating cycle from 120 to −60 °C and again to 120 °C at −20 °C min^−1^.

Conductivity was determined in a NOVOCONTROL GmbH Concept 40 Broadband Dielectric Spectrometer (Novocontrol Technologies, Montabaur, Germany) in the temperature range of −50 to 90 °C, in a frequency range between 0.1 and 107 Hz. Disk films of dimensions of 2 cm diameter and ~500 μm thickness were sandwiched between two gold-plated flat electrodes. A frequency sweep was done every 10 °C from −50 to 90 °C. Thereafter the same measurements were done, but cooling from 85 to 25 °C. The ionic conductivity (σ) of the samples was calculated by using the conventional methods based on the Nyquist diagram and the phase angle as a function of the frequency plot [10].

Self-creep tests were done on electrolyte films about 500 μm thick, which were sandwiched between gold-plated flat electrodes of 20 mm diameter. Then the sandwich was placed on a heating plate with 0.5 kg on top at 70, 90, and 100 °C for 20 min at each temperature. More details appear in [11,12]. The electrolytes that maintained their shape at all three temperatures were considered solid-like.

Profilometry was carried out using a Zeta Instruments Zeta-20 profilometer at 5x, 20x, and 50x magnification. 

## 3. Results

Though it is possible to adapt a melt compounder to work under a controlled atmosphere, for practical reasons it is more prudent to design a two-step method as that described in the experimental section of this work. This two-step procedure has the advantage of allowing very flexible methodologies for the specific requirements of the different electrochemical phases. It basically consists of separating the preparation of the mechanically active phase, i.e., the one that will confer solid-like behavior to the final electrolyte, from the incorporation of the electroactive liquid phase, which can be done in a controlled atmosphere and even in a glovebox. As illustrated in the photographs of Scheme 1, both procedures produce self-standing films, homogeneous to the naked eye, flexible, and easy to handle. This methodology can be used with other polymers different from PEO, it only requires the polymer to be swollen by the liquid phase and for the final membrane to behave as a solid. These badly mixed electrolytes will be microscopically heterogeneous, with polymer-rich and polymer-poor phases, a morphology which may become advantageous for ionic conductivity. It is especially interesting for polymers that behave as cationic conductors, like PEO, because cationic conduction can then take place though the liquid-rich phase, through the interphases between the polymer-rich and the liquid-rich phase, and even (though more slowly) through the polymer-rich phase itself. 

For the preparation of Li electrolytes with RTIL, choosing liquids containing the pyrrolidinium cation rather than the imidazolium one would seem wiser because of its higher electrochemical stability [22]. However, the purpose of this work is to check the effect on ionic conductivity of a controlled phase separation morphology and to prove the validity of the two-step procedure to prepare electrolytes with humidity sensitive electroactive phases, and in particular, for aluminium batteries, in which we are interested, where imidazolium chloride is frequently used.

### 3.1. Composition Heterogeneity of EMITFSI-2 as Compared to EMITFSI-1

Figure 1a collects the characterization of EMITFSI-1 and EMITFSI-2 membrane surfaces by ATR–FTIR, together with pictures of the membranes. Both electrolytes are self-standing and to the naked eye, they look very similar. In EMITFSI-1 a few TPGS-S aggregates can be seen, while in EMITFSI-2, where TPGS-S seems better dispersed, opaque heterogeneities on the surface appear. ATR–FTIR is performed at different places on the EMITFSI-1 and EMITFSI-2 surfaces. ATR–FTIR in EMITFSI-2 reveals a strong heterogeneity in composition, with regions richer in PEO and others richer in EMITFSI/LiTFSI. FTIR spectra of different regions appear collected in Figure 1a. Polymer-rich regions are characterized by lower intensities of the band at 1180 cm^−1^, characteristic of EMITFSI, compared to the band at 1280 cm^−1^, characteristic of PEO, while the opposite occurs for the polymer-poor regions. Regions which are visually more opaque are polymer-rich, while less opaque regions are polymer-poor. This is seen throughout the film, the composition of which can be considered to be heterogeneous at this macro/microscopic length scale. The same occurs when recording ATR–FTIR on the surface of EMITFSI-2a (Figure 1, in orange) or EMITFSI-2b (not shown). On its turn, ATR–FTIR proves that the composition of EMITFSI-1 is homogeneous at the length scale of the ATR–FTIR throughout the membrane surface, and only one representative FTIR spectra has been included in Figure 1. 

Figure 1b shows 2D images of EMITFSI-2 obtained by profilometry at 50x. These images have been taken choosing membrane regions which appeared more or less opaque to the naked eye, though these are difficult to see because their size does not exceed a few millimeters. Two different textures are seen in this electrolyte’s surface; one is smoother, corresponding to the liquid-rich phase (left), and the other rougher (right) and belongs to the polymer-rich phase. 

### 3.2. Relaxations, Transitions, and Phase Morphology as Studied by DSC

Figure 2 presents the DSC traces of the first cooling scan (Figure 2a) and the second heating scan (Figure 2b) of neat PEO, PEO melt compounded with a 5 wt % of TPGS-S, and the electrolytes EMITFSI-1, EMITFSI-2, EMITFSI-2a, and EMITFSI-2b. The *T*_g_ and *T*_m_ of the electrolytes are well seen in the DSC scans and their values appear in Table 2. Figure 2a shows the typical crystallization exotherm of PEO at about 45 °C, while the nucleating effect of TPGS-S makes the blend of PEO with a 5 wt % of TPGS-S crystallize at about 50 °C. The sample PEO/TPGS-S is the polymeric scaffold prepared in Step 1 of the two-step procedure. Crystallization is impeded in the four electrolytes, and almost no crystallization exotherm appears in the cooling scan. This is normal behavior in blends of PEO with large fractions of EMITFSI [15]. In EMITFSI-2b, which has no LiTFSI nor TPGS-S, a small exotherm is seen at about 0 °C, barely seen as well in EMITFSI-2a (in expanded scale), without TPGS-S. Both EMITFSI-1 and EMITFSI-2 show similarly flat cooling scans, i.e., no PEO crystallization under those cooling conditions, which suggests that the swelling of PEO in EMITFSI-2 is complete. 

EMITFSI on its own melts at −16 °C, and solutions of LiTFSI in EMITFSI may also present melting peaks (broader than pure EMITFSIs) in that region, depending on the concentration. It is noteworthy then that no crystallization of the EMITFSI + LiTFSI liquid phase is seen either, which also indicates that although in EMITFSI-2 a microscopic phase separation exists, as characterized by ATR–FTIR, there is a high degree of mixing of the components, so that no pure liquid or pure polymer phases can be characterized by thermal properties.

Figure 2b shows the second heating scan of all the samples under study. The *T*_g_ of PEO and PEO + 5 wt % TPGS-S appears as a broad transition at about −54 and −52 °C, respectively. Electrolytes EMITFSI-1 and EMITFSI-2 show a narrower and larger *T*_g_ at about −51 and −53 °C, i.e, the same temperature as in pure PEO. The *T*_g_ in the electrolytes is larger than in PEO because they are less crystalline, and so contain a larger fraction of amorphous phase. Narrow *T*_g_ indicates homogeneous domains with regards to the onset of segmental relaxation, which again suggests that the PEO scaffold in EMITFSI-2 is made to be very swollen by the EMITFSI + LiTFSI solution, as in EMITFSI-1. 

EMITFSI-2a, with no TPGS-S but with LiTFSI, also has a sharp *T*_g_ at about −50 °C, i.e., very similar to EMITFSI-1 and EMITFSI-2. However, EMITFSI-2b, with no LiTFSI, has a *T*_g_ at a significantly lower temperature, about −60 °C. These differences in *T*_g_ illustrate the plasticization of PEO chains by EMITFSI, which produces the *T*_g_ decrease in EMITFSI-2b.

The melting region in Figure 2b shows two strong endotherms for PEO and PEO + 5 wt % TPGS-S, both melting at about 65 °C. On their turn, the four electrolytes show very small (EMITFSI-1, EMITFSI-2a, and EMITFSI-2b) or even absent (EMITFSI-2) endotherms, in accordance with the cooling scans in Figure 2a. EMITFSI-1 (as well as EMITFSI-2a and EMITFSI-2b) are almost completely liquid, though they include a small fraction of crystalline phase which disappears at *T* > 40 °C.

EMITFSI-2 is, according to the thermograms in Figure 2, a liquid and not a solid material at *T* > −50 °C. There is no crystalline phase at all, and the glass transition takes place at *T* = −53 °C. So strictly speaking, this material is a phase-separated liquid where a polymer-poor (or EMITFSI-/LiTFSI-rich) phase of lower viscosity coexists with a polymer-rich phase of higher viscosity. These two phases are highly intermingled, connected by large interfacial regions, to the point that the thermal fingerprint (*T*_m_ and/or *T*_c_) of the pure components is completely absent from the thermograms. In this sense, these phase-separated electrolytes are a completely different concept to soaked porous membranes or tissue-like soaked separators, firstly, in that the electrolytes described in this work are dense and not porous membranes, and secondly, in that no domains of pure polymer or pure liquid exist. Scheme 2 illustrates the three types of morphologies: Porous, homogeneous, dense (like EMITFSI-1), and phase-segregated dense membranes (like EMITFSI-2). Though both porous and dense electrolytes may be dimensionally stable and display other solid-like characteristics at the macroscopic scale, their morphology and structure at the micro and nano scales are completely different in ways that may be very relevant for the development of performing electrolytes. In particular, properties that depend on the structure at the nano and microlevel will be very different. Ion diffusion is among the properties related to the nano and microstructure, and ion diffusion governs not only ionic conductivity but also metal dendrite growth. 

### 3.3. Rheology and Conductivity

In previous work [12,13,14,15], we showed how the incorporation of the modified sepiolite fiber TPGS-S produces a physical crosslinking with the PEO chains that makes electrolytes like EMITFSI-1 behave as solid thermoplastics even when almost no crystalline phase of PEO exists. These electrolytes can in fact be described as physically crosslinked gels, as they contain a very large liquid fraction of liquid phase. In addition to complete rheological studies [12,13,14,15], a simple self-creep test, such as that described in the experimental section of this work, is used to prove their solid-like behavior from room temperature to 90 °C. The question is whether the electrolyte EMITFSI-2, which was done by swelling the polymeric scaffold with the liquid phase EMITFSI + LiTFSI, will also display a solid-like behavior and be able to endure the self-creep test. 

Figure 3 shows images of the result of the creep test in electrolytes EMITFSI-1, EMITFSI-2, and EMITFSI-2a, the latter without TPGS-S. The creep test employed in this work was very demanding, and included a final 20 min stage at 100 °C. While EMITFSI-1 and EMITFSI-2 withstood the test, EMITFSI-2a softened and flowed, proving once again the great performance of TPGS-S as a physical crosslinker of PEO. Though not shown in the figure, EMITFSI-2b, which had no TPGS-S nor LiTFSI, flowed like EMITFSI-2a. The physical crosslinking of PEO by TPGS-S was described in detail before [12,13,14,15], but it is the first time being seen occurring in the absence of melt-compounding blending of the polymer and the liquid phase. The excellent behavior of EMITFSI-2 in the creep test is yet another indication of the complete intermingling of the PEO-rich and PEO-poor microphases in this electrolyte, for solid-like behavior of the membrane would otherwise not be possible.

The σ of the electrolytes EMITFSI-1, EMITFSI-2, and EMITFSI-2a was measured from −50 °C to 90 °C, on heating and on cooling. The results appear in Figure 4a,b, and in Table 2. Figure 4a shows how the σ of EMITFSI-2 increases with T in a continuous way, without sigmoidal-like variations. This is in strong agreement with the DSC traces in Figure 2, which shows that this electrolyte has neither crystallization nor melting from −70 °C to 100 °C. EMITFSI-1 displays lower σ than EMITFSI-2 in all of the T range, and the difference is larger at 40 °C than over: In the range of −50 °C to 20 °C, the σ of EMITFSI-2 just about doubles that of EMITFSI-1, while over 40 °C σ, it is about 20% higher in EMITFSI-2. This is so because the melting of the scarce crystalline phase in EMITFSI-1 at about 40 °C (see DSC traces in Figure 2) produces a mild sigmoidal-like increase in conductivity, which occurs between 20 °C and 40 °C. The same occurs with electrolyte EMITFSI-2a.

EMITFSI-2a (not shown on cooling) is very similar to EMITFSI-1 and its σ suffers sigmoidal step-like increases in the vicinity of the melting of the PEO crystallites, between 20 and 50 °C, also in good accordance with the DSC of this electrolyte, shown in Figure 2. A consequence of these phase transitions is that the σ values measured at the same temperature, but on heating from low temperatures (under crystallization processes) or cooling from high temperatures (over the melting temperature), are not the same. This is seen in Figure 4b, where σ measured on cooling appears as open symbols. In EMITFSI-2 measurements done on cooling and on heating are very similar, as it corresponds to an electrolyte without phase transitions in the temperature range, but in EMITFSI-2a and EMITFSI-1 σ, measured on cooling is significantly higher, and the difference between σ-cooling and σ-heating becomes larger as the measurement temperature approaches the crystallization (on cooling) or melting (on heating) temperature of the electrolyte. 

The actual σ values of electrolytes EMITFSI-1 and EMITFSI-2 are very large, especially bearing in mind that they are solids up to 90 °C. Values of about 0.5 mS/cm and 4 mS/cm are obtained for these electrolytes at 25 and 70 °C, respectively (see Table 2), which are very remarkable by themselves. The data in Table 2 also show that σ is about 20% higher in EMITFSI-2, compared to EMITFSI-1, both at 25 and 70 °C. This proves that inducing a microphase separation by using the two-step procedure proposed in this work does make σ of these thermoplastic electrolytes higher.

## 4. Conclusions

Electrolytes comprising PEO as a polymeric scaffold and a solution of EMI TFSI and Li TFSI as liquid phase were prepared using a solvent-free method that allows thermoplastic solid electrolytes to be obtained in controlled atmosphere conditions. This two-step method consisted of a first stage, where PEO was melt-compounded with TPGS-S, a physical crosslinker of the polymer, and pelletized, and a second stage where the pellets were swollen with the LiTFSI/EMITFSI liquid phase. This second step was done in controlled atmosphere conditions, so electroactive species that are very sensitive to atmospheric conditions, for instance, some Li and Al electrolytes, could be used. The two-step method resulted in an electrolyte with controlled phase-segregated morphology, i.e., polymer-poor and polymer-rich phases, with increased ionic conductivity over that of homogeneous electrolytes and comparable mechanical stability. The concept developed in this work was therefore successful and there is room for increasing σ even more in phase-segregated electrolytes by further optimization of the second step of the procedure. This second step would determine the electrolytes’ final morphology, i.e., the size, distribution, intermingling, and tortuosity of the polymer-poor and polymer-rich phases, and these morphological features would control the electrolytes’ rheology and σ. Compared to the single step melt-compounding methodology, this new method i) is adapted for liquid phases that have to be handled under a controlled atmosphere, like aluminium electrolytes, ii) permits the tuning of microphase segregation, thus increasing σ, and iii) is still solvent-free and scalable like the one-step procedure.

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
