# Peer review of "Solvent-Free Procedure for the Preparation under Controlled Atmosphere Conditions of Phase-Segregated Thermoplastic Polymer Electrolytes"

_polymers, 2019, doi:10.3390/polym11030406_

Round 1

Reviewer 1 Report

This manuscript describes the preparation of liquid ions containing PEO polymer electrolyte membranes by a solution-swelling method. The resulted membranes were characterized by FTIR, DSC and dielectric spectroscopy. However, as a reader, the scientific soundness of this paper seems low for Polymers. The abstract should be rewritten, and the conclusion should be given. The title (Solvent-free procedure ----under controlled atmosphere conditions) is not appropriate for the content.

Author Response

Answer to Reviwere 1 has been uploaded

Reviewer 2 Report

The authors report a novel preparation method for polymer electrolytes on the basis of poly(ethylene oxide), lithium salt, and ionic liquids. In particular, their two-step method (incorporating a filler into a PEO melt which is subsequently doped with liquid electrolyte) allows the fabrication of membranes with alternating regions that are locally enriched by the liquid electrolyte or the polymer, respectively. This results in an enhanced mechanical stability, and, to a minor extend, an increased conductivity.

Overall, I find no flaws with the presented results, and therefore recommend the publication of the paper.

Author Response

Reviewer 2 has well understood the purpose and results of the work.

Reviewer 3 Report

The manuscript submitted by Miguel et al. reports a solvent-free, two-step preparation method of poly(ethylene oxide) (PEO) based electrolytes in rechargeable batteries. The aim of this work is to synthesize thermoplastic electrolytes with polymer-ionic liquid separated phases having high ionic conductivity and suppressing metal dendrite growth.

The topic of the submitted work matches well with the journal scope. The topic of polymer electrolytes for rechargeable batteries is critical for the advancement of electrochemical energy storage, therefore I expect this work to be impactful. The strength of this work is its introduction which describes clearly the background and the motivation; however, the results and discussion section needs elaboration for clarity. This work could be accepted pending a satisfactory revision to fully address the following comments.

1. Line 188-189: Please supply one or more references for the electrochemical stability of pyrrolidinium cation.

2. In the Introduction section, please briefly explain the rationale of choosing sepiolite as the additive.

3. Figure 1:

1) Please label panels a and b.

2) Please provide FTIR spectra and profilometry 2D images of EMITFSI-1, EMITFSI-2a, and EMITFSI-2b, along with corresponding discussions as well.

4. Line 231: What does "can also be suspected" mean?

5. Figure 2: the y-axes are ill defined. Please give the exact names and units. Label whether endothermic peaks point up or down.

6. Line 263, "...the plasticization of PEO chains by EMITFSI, which produces the Tg (Tg) decrease in EMITFSI-2b": Because EMITFSI does not contain EMITFSI, it is invalid to correlate the influence of EMITFSI to EMITFSI-2b.

7. It is necessary to provide the creep test result of EMITFSI-2b in Figure 3.

8. Figure 4(b): the unit of the x-axis is not correctly shown.

9. Please add the discussion on the ionic conductivity difference between heating and cooling processes (Figure 4b).

10. The ionic conductivities of EMITFSI-2a and EMITFSI-2b during heating and cooling are missing.

Author Response

Answer to Reviewer 3 has been uploaded

Round 2

Reviewer 1 Report

The manuscript has been well modified, and the other two references give the positive comments. So I think it can be accepted in previous versions.

Author Response

I am uploading the final revised version

Reviewer 3 Report

The authors have provided an extensive revision which addresses all my original comments. Some of the responses from the authors are great and wish the authors could incorporate them into the manuscript to enhance the overall presentation clarity. Specifically:

Some data can be included in a Supporting Information file if the authors concern that adding such data will compromise the illustration clarity. These data include: 1) the creep test result of EMITFSI-2b (response #7); 2) the ionic conductivity-temperature profile of EMITFSI-2a (response #9).

Response to Comment #6: Although the authors mentioned they did not understand the comment, I checked the revised manuscript and found the sentence which I questioned were properly revised. Therefore, I have no further comment.

Response to Comment #8: I encourage the authors to check the submitted version rather than their own copy. The units of the x-axis are still incorrectly shown, possibly due to compatibility issue.

Response to Comment #10: It is great that the authors mentioned it was impossible to test the impedance of EMITFSI-2b. I hope this information can be included in the revised manuscript.

Author Response

The ionic conductivity profile has been included in figure 4

Now I see that the degrees centigrade units in figure 4 are in chinese but only in the pdf version, not in the word version so I suppose that is OK.

We have included a sentence saying that it is not possible to measure ionic conductivity as a function of T in EMITFSI-2b